# Composition and Clinical Significance of Exosomes in Tuberculosis: A Systematic Literature Review

**DOI:** 10.3390/jcm10010145

**Published:** 2021-01-04

**Authors:** Fantahun Biadglegne, Brigitte König, Arne C. Rodloff, Anca Dorhoi, Ulrich Sack

**Affiliations:** 1College of Medicine and Health Sciences, Bahir Dar University, 79 Bahir Dar, Ethiopia; 2Institute of Medical Microbiology and Epidemiology of Infectious Diseases, Medical Faculty, University of Leipzig, 04103 Leipzig, Germany; Brigitte.Koenig@medizin.uni-leipzig.de (B.K.); acr@medizin.uni-leipzig.de (A.C.R.); 3Institute of Clinical Immunology, Medical Faculty, University of Leipzig, 04103 Leipzig, Germany; Ulrich.Sack@medizin.uni-leipzig.de; 4Friedrich-Loeffler-Institut, 17493 Greifswald-Insel Riems, Germany; Anca.Dorhoi@fli.de

**Keywords:** exosomes, isolation, exosomal markers, protein, miRNA, tuberculosis

## Abstract

Tuberculosis (TB) remains a major health issue worldwide. In order to contain TB infections, improved vaccines as well as accurate and reliable diagnostic tools are desirable. Exosomes are employed for the diagnosis of various diseases. At present, research on exosomes in TB is still at the preliminary stage. Recent studies have described isolation and characterization of *Mycobacterium tuberculosis* (*Mtb*) derived exosomes in vivo and in vitro. *Mtb*-derived exosomes (*Mtb*exo) may be critical for TB pathogenesis by delivering mycobacterial-derived components to the recipient cells. Proteomic and transcriptomic analysis of *Mtb*exo have revealed a variety of proteins and miRNA, which are utilized by the TB bacteria for pathogenesis. Exosomes have been isolated in body fluids, are amenable for fast detection, and could contribute as diagnostic or prognostic biomarker to disease control. Extraction of exosomes from biological fluids is essential for the exosome research and requires careful standardization for TB. In this review, we summarized the different studies on *Mtb*exo molecules, including protein and miRNA and the methods used to detect exosomes in biological fluids and cell culture supernatants. Thus, the detection of *Mtb*exo molecules in biological fluids may have a potential to expedite the diagnosis of TB infection. Moreover, the analysis of *Mtb*exo may generate new aspects in vaccine development.

## 1. Introduction

According to the world health organization (WHO), tuberculosis (TB) is a major health problem in developing countries. The WHO report [1] indicated that about a quarter of the world’s population is latently infected with *Mycobacterium tuberculosis* (*Mtb)*, but only 5–10% of the exposed individuals fall ill, develop active TB, and will subsequently infect others. The Sustainable Development Goals created by the UN set 2030 as a global target to end the TB epidemic [2]. However, the eradication of TB has been troublesome partly due to the ability of the TB causing bacteria to survive latently for years in the host without causing active diseases. Latent TB means that bacilli persists within the infected host and the adaptive immune system responds to stimulation with mycobacterial antigens [3]. *Mycobactrium tuberculosis* (*Mtb)* infects primarily macrophages which may destroy the pathogen upon activation or eventually process it into short-peptide fragments for antigen presentation [4]. Thus, to contain TB, a vaccine and accurate, reliable diagnostic tools must be continually updated to address TB Infection Prevention and Control measures (IPC).

Exosomes are small-sized particles that are produced by the inward budding of early endosomes [4]. Within most cells such budding membranes mature into multivesicular bodies (MVBs) and fuse with the plasma membrane thereby mediating secretion of exosomes [5,6]. Early endosomes, originated during endocytosis of the plasma membrane, and MVBs, are involved in the internalization and transport of extracellular material within the cell [7]. In the process of exosome release, MVBs could be either degraded by the lysosome or fused with the cell membrane [5]. The precise fate of MVBs during the process is not well understood [8]. Exosomes are different in origin, size, and content and involved in the pathogenesis of infection [9]. Studies performed over the past years proved that exosomes exist in almost all biological fluids [10,11], including plasma [12,13], serum [14], bovine and human milk [15], urine [16], and saliva [17]. Importantly, an increasing evidence suggests that exosomes play critical role in cell-to-cell communication [18,19], including the interaction between microorganisms and infected human cells. Exosomes modulate host immune and inflammatory responses during infection [20,21], and have been also suggested to be applicable in basic research, disease diagnostic biomarkers and vaccine discovery [4,22], and in therapeutic targets [23,24]. Exosomes application as a liquid biopsy in the field of oncology, pregnancy disorders, cardiovascular diseases, and organ transplantaion has been documented [25]. In addition, they have been suggested to be amenable as an effective liquid biopsy technique in non-invasive tumor diagnosis and monitoring [26,27,28,29], and are useful to evaluate the therapy response in disease [30].

One major challenge in TB is to improve and standardize simple, cost effective diagnostics, devise reliable and accurate treatment and therapy monitoring programs and design better vaccines. Exosomes are one type of bioactive vesicle that has received considerable attention in TB research. This review focuses on and discusses the different exosome analysis technique platforms applied in TB, molecular contents of exosomes, and their clinical significance in TB infection.

## 2. Methods

A literature search was conducted for articles published using the online databases PubMed/Medline and Google scholar. To find articles related to TB and extracellular vesicles and/or exosomes, we considered one of the following key words: “*Mycobacterium tuberculosis*” and “culture and biological fluids”, “extracellular vesicles/exosomes”, “isolation and characterization”, “proteomics”, “protein and miRNA cargo”, “DNA”, and year of study. Each term was searched separately. Only English-language papers and International Society for Extracellular Vesicles (ISEV) websites were included in the search, the searches were focused on the studies of isolation, and characterization, proteomics, miRNA analysis of *Mtb* derived EVs/exosomes, culture, and biological fluids. Only study reports that used original research papers were reviewed. To minimize selection bias, all possible relevant articles were evaluated critically and those that met the inclusion criteria were selected. Literature that did not report on a study of *Mtb* derived exosomes was excluded. A structured form for data extraction was used and included details such as time and place where the study was conducted, year of publication, method of isolation of exosomes, detected exo-protein, exo-miRNA and/or exo-DNA of *Mtb*. We assessed and recorded exosomal markers associated to TB frequency across all studies reported (Figure 1).

## 3. Review

### 3.1. Finding

A variety of research papers published in the extracellular vesicles and/or exosomes were different in terms of research methods, purposes, and study site. As summarized in Figure 1, our electronic search resulted in 2240 citations. Finally, 118 articles were retrieved for full text review and 53 papers were determined to meet eligibility criteria and included in this review. Exosomes as diagnostic markers in TB is the most reported followed by vaccine candidate, indicating that diagnostics and vaccine discovery has paid great attention in TB exosome research. The summary of studies by conculusion and clinical significance of exosomes in TB is presented in Figure 2. Among the 53 published papers on TB exosome research, 25 were on exosomal-proteins (exo-proteins) and followed by exosomal miRNAs (exo-miRNAs) (Figure 3). The studies used proteomics and western blot techniques for the identification of exo-proteins (Figure 4). The Nano sight NS300 and western blot characterization of exosomes extracted from THP-1 cell culture supernatants is presented in Figure 5. 

### 3.2. Mtb Exosomes Isolation and Characterization Methods 

To date there are a variety of approaches to isolate exosomes effectively from biological fluids and cell culture supernatants. However, the methods differ in yield, purity, and size distribution of isolated exosomes [22]. Exosome-isolation techniques that are commonly available in infectious and noninfectious diseases can be divided into ultracentrifugation, density-gradient centrifugation, ultrafiltration, precipitation, and immunoisolation [31,32,33]. This section discusses the different types of techniques to isolate exosomes in TB. Isolation methods affect the purity and yield of exosomes [34]. A variety of specimens can possess different exosomal contents which requires suitable isolation methods [35]. For example, if exosomes are to be isolated from cell culture media, either serum-free media or exosome-free fetal bovine serum should be used. 

Low-speed centrifugation removes cells, apoptotic debris, and larger vesicles (10,000× *g*) and a high-speed centrifugation enriches exosomes (100,000× *g*). The most widely used method in TB exosome research is differential centrifugation followed by ultracentrifugation. Palacios A et al. [36] and Lyu L et al. [37] in 2019 extracted *Mtbexo* using ultracentrifugation from culture supernatants and serum, respectively. Other studies have been conducted using the ultracentrifugation method for the extraction of *Mtb* related exosomes from cell culture supernatants [38,39,40] and biofluids [41,42]. However, damage of exosomes due to high speed and large volume sample are the major drawbacks of this method [43]. Furthermore, the viscosity of the biological fluids affects the purity of exosomes [44], indicating that samples with high viscosity require longer ultracentrifugation and a higher speed of centrifugation.

A form of centrifugation that is used to separate exosomes based on their buoyant features in a sucrose gradient is density gradient centrifugation [45]. The method combines ultracentrifugation with sucrose density gradient [46]. Athman JJ and his colleagues in 2015 isolated *Mtb*exo by density gradient centrifugation [47]. Furthermore, SinghPP et al. in 2011 isolated exosomes from *Mtb*-infected and uninfected RAW264.7 cells using sucrose density gradient [38]. The method is used to separate exosomes from proteins and aggregates of different densities and facilitates isolation of exosomes from biological fluids [33]. Another fast and simple technique used to isolate exosomes based on pore appropriate size is size-exclusion chromatography (SEC) [35,48]. The method applies a column packed with porous polymeric beads containing multiple pores and tunnels [33], allowing the molecules to pass through the beads depending on their size. A study conducted by Garcia-Martinez M et al. in 2019 used SEC to isolate EVs from J774A.1 macrophages [49]. Commercial kits to isolate exosomes, particularly polyethylene glycol and proprietary chemicals precipitate exosomes. The method is time saving, easy and user-friendly; however, there is a considerable risk of co-precipitation of plasma proteins. Tyagi and his colleagues used ExoQuick Ultra (System Biosciences, Inc, CA) and 0.22 µm pore size filters for the isolation of exosomes from *Mtb* infected cell culture supernatants [40], and the total exosome isolation kit (Invitrogen) [50] were used by Dahiya B et al. for the isolation of exosomes from urine of TB patients. The interaction between exosomes surface receptors and antibodies remained a matter of choice in exosome research. Antibodies specific to surface proteins of EVs (e.g., CD9, CD81, CD63, TSG101, and Alix) are linked to chemically modified or protein-coated beads, and capture exosomes by binding to these proteins [51]. The characteristics of exosomes isolated based on immunoaffinity differ from those isolated based on the physicochemical characteristics of the exosomes (e.g., size, surface potential, and density). The physicochemical-based separation methods cannot distinguish between different cellular origins [35,48]. 

There is no consensus in the scientific community regarding which method to be used for the characterization of exosomes. However, following isolation and purification a variety of methods can be implemented for exosome characterization in terms of size, quantity, and morphology. These methods include biophysical, molecular, and microfluidic techniques [33], transmission electron microscopy (TEM), nano sight tracking analysis (NTA), Raman spectroscopy (RS), and atomic force microscopy [52,53,54,55,56], respectively, and field-flow fractionation [57], photon correlation spectroscopy [58], Western blot (WB) [59], and/or ExoCarta database [60]. NTA measures the concentration and size distribution of nano-sized vesicles. Furthermore, the shape of the purified exosomes can be visualized by TEM. The typical shape of exosomes is spherical, approximately 30–150 nm [61]. Smith VL et al. [62] and Athman JJ and his research team [47] implemented both NTA and TEM for the characterization of *Mtb*exo. Furthermore, Giri PK et al. charactrerized *Mtb* exosomes using electron microscopy [63]. Wiklander and his colleagues commonly used flow cytometry to characterize EVs subpopulations in cells, a method described in detail in 2018 [64]. Exosomes released from *Mtb*-infected cells were also characterized by flow cytometry [38]. Several methods have been developed for analysis of exosomal RNA content [65,66,67]. With regard to proteins, the protein content of exosomes is analyzed by western blotting, proteomic technology, and fluorescence-based cell sorting [63,68,69]. We characterized and evaluated the size distribution and the concentration of exosomes extracted from naïve and *Mtb*-infected THP-1 cell culture supernatants using Nano sight NS300 (Nano sight Ltd, Amesbury, UK) and detected specific surface protein markers (CD9, CD63, and CD81) [unpublished data] (Figure 5).

### 3.3. Composition of Host Exosomal Surface Markers by Mtb Infected Cells

Cells release exosomes under physiological and pathological conditions [6], which contain various membrane and cytosolic proteins. The biogenesis of exosomes and transportation of MVBs are regulated by the endosomal sorting complexes required for transport (ESCRT) proteins [69]. Regardless of the type of the cell from which the exosome originate ESCRT proteins and proteins such as Alix, TSG101, HSC70, and HSP90β are components of exosomes [32]. The molecular content of exosomes varies with their cells of origin. According to the ExoCarta, a database of exosomal proteins, RNA, and lipids developed by Mathivanan and Simpson in 2009 [60], 4563 proteins, 194 lipids, 1639 mRNAs, and 764 miRNAs have been recognized in exosomes. The membrane transport proteins and fusion proteins are most frequently detected. The CD63 along with CD9 and CD81 are proteins of the tetraspanins family [59]. These transmembrane exosomal proteins such as tetraspanins (CD9, CD63, and CD81), heat shock proteins (HSC70 and HSC90) are considered exosomal marker proteins [60,69]. According to Tickner and his colleagues, exosomes are rich in lipids such as cholesterol, phospholipids, phosphatidylserine, and prostaglandins. In addition, exosomes contain nuclear, mitochondrial, and ribosomal proteins [70]. Diaz G and his colleagues reported CD63, CD81 and HSP70 from *Mtb* infected and uninfected cells [71]. The exosomal markers CD63, CD9, and CD81 along with GAPDH were isolated from exosomes of *Mtb* infected and näive macrophages [47]. Exosomes isolated from *Mtb* infected cells were found to contain both MHC-I and II [72], suggesting that the vesicles are capable of inducing an antigen specific T cell response. Similar reports by Giri PK and Schorey JS in 2008 showed the expression of CD86, MHC-II and CD83 in *Mtb* infected cells. Another study that employed enzyme-linked immunosorbent assay (ELISA) reported host HSP25, HSP60, HSP70, and HSP90 proteins in the serum samples of TB patients [73]. A different study reported host proteins such as LAMP-1 [62,74] and teraspanins like CD9, CD63, CD81, and CD86 [75]. The composition of host exosomal surface markers by *Mtb* infected cells is summarized in Table 1.

### 3.4. Mtb Molecules Extracted from Exosomes 

Exosomes have been suggested to be applicable in TB diagnostics and vaccine innovation [4,47,50,62,63,71,73,82,84,85,86,87,88,89]. Proteomics analysis of exosomes released by *Mycobacterium*-infected cells have indicated the presence of lipoproteins [39], some of which were reported as a virulence factors. Furthermore, Prados-Rosales R et al. detected DnaK, 19 kDa (Lpqh), and LprG and LAM by immunoblot analysis. The study also confirmed EVs by TEM as a vesicle associated morphology. In the report, lipoproteins 19 kDa and LprG were located at the vesicle membrane, nevertheless, DnaK localized to the lumen of the vesicles. The findings of Smith VL and his collegues in 2017 imply that exosomes function to promote T-cell immunity during TB infection and are an important source of extra-cellular antigen [62]. There is a growing evidence that proteins and miRNA secreted by *Mtb* are enclosed within exosomes [63,84,90], and transfered into the extracellular environment. 

Studies suggests that TB bacteria may use exosomes as a mechanism of releasing molecules into host cells [88,89,91], which could be therefore, considered as potential virulence factors involved in the pathogenesis of TB. Another proteomic analysis of exosomes revealed broad range of proteins within the mycobacterial exosomes [63,86]. Giri and his colleagues in 2010 reported 29 *Mtb* associated proteins from J774 cells treated with *Mtb* culture filtrate proteins (CFP) [63]. In a similar study, Giri and his team also reported Ag85a,b,c, KatG, Mpt63, Mpt64, HspX, Mpt53, TB22.2, GroES, Mpt51, BfrB, and ESAT-6 from *Mtb* treated cells [63]. Kruh-Garcia et al. detected twenty proteins from serum in the exosomes of active TB patients, and included multiple peptides from eight proteins (Ag85 b,c, Apa, BfrB, GlcB, HspX, KatG, and Mpt64), in which Ag85b,c, Apa and GlcB proteins are known mycobacterial adhesions [86] and BfrB, HSPx, katG, and MPt64 [92,93] contribute to the intracellular survival of *Mtb*. Furthermore, Fortune SM and colleagues isolated and evaluated a high concentration of Ag85a, HSPx, and K85R in a mouse model of *Mtb* infection [94]. Shekhawat SD et al. in 2016 reported significantly up-regulated exosomal heat shock proteins (HSP-90, HSP60, and HSP70) in serum of latent TB patients. This suggests that HSPs could be detected in the serum quickly, and could be potentially used for the rapid diagnosis of latent TB, particularly for monitoring TB infection in areas where TB is rampant [73]. Exosomes secreted from *Mtb-*infected macrophages contain molecular components of TB bacteria including 19-kDa lipoarabinomannan, thereby conveying immunologic information among macrophages and modulating production of TNF-α and other cytokines that boost anti-mycobacterial activities of the macrophages [39,74]. The studies indicated that host-derived exosomes carry specific proteins of TB bacteria, thus exosomes with specific mycobacterial antigens may activate non-specific or specific immune responses in host cells to eliminate *Mtb* [62]. Another study reported exosomal TB related markers such as Ag85A, Ag85B, Ag85 C, MPT64, and LprG [95,96]. Dahiya B et al. in 2019 isolated and evaluated *Mtb* related exosomal CFP10 and lipoarabinomannan from urine of TB patients [50]. The *Mtb* related protein molecules Ag85 a, HSPx, and K85R were reported as an important antigens during *Mtb* infection [62]. Rosenkrands I et al. reported *Mtb* related proteins including TB22.2, ESAT-6, PrcB, KatG, HSPx, and Ald [90]. According to the bioinformatics analysis by Singhal et al. in 2012, proteins encoded by Rv0635, Rv1827, and Rv0036c and Rv2032, Rv2896c, and Rv2558c genes are involved in the cellular metabolism for the intracellular survival of *Mtb* [97]. A study by Castro-Garza and colleagues detected HSPx protein in sera of TB patients, suggesting the identification of recent latent TB infection for TB prevention and control strategy [98]. Furthermore, Mehaffy C et al. in 2020 detected HSPx in sera of latent TB infection [88]. Another study reported *Mtb* H37Rv culture filtrate proteins including the most components of immunologically important ESAT-6 family [84]. Previous studies also reported *Mycobacterium* vesicles associated proteins such as MPT-51, Ag85C, MPT32, and an 88-kDa protein [99], which have diagnostic potential, in serum of TB patients. In addition, Samanich, K.M et al. reported Ag85C, MPT32, and the 88-kDa protein in sera of TB patients [99]. Wanchu and his research team identified MS and MPT51 proteins in TB patients [100]. *Mtb* molecules extracted from exosomes are summarized in Table 2.

Exosomal RNAs have been studied in diverse diseases but have been limited in mycobacterial infection [103]. The isolation and evaluation of miRNA in TB has received increased attention in context of TB diagnosis. Progression of active TB, i.e., subsequent interactions between *Mtb* and host cells, is regulated by miRNAs [79]. These miRNAs are involved in several key metabolic pathways including central carbon metabolism, fatty acids and sugar metabolism, amino acid metabolism, bacterial invasion related pathways, and cell signaling pathways [77]. Multiple host pathways are implicated in immune surveillance and are modulated to enable bacterial survival within infected macrophages. The infection of human macrophages with the virulent *Mtb* H37Rv results in a distinct pattern of miRNA expression [103]. Singh PP et al. identified 57 miRNAs in exosomes released from infected macrophages including Mmu 223 and 486-5p [103]. Singh Y et al. [104] documented that miR-99b is highly upregulated in *Mtb* infected dendritic cells (DCs). This study indicated that miR-99b targets TNF-α and TNFRSF-4 receptor gene transcripts. Nevertheless, the less abundance of this miRNA leads to a significant loss in bacterial survival in DCs. The up-regulation of miRNA miR-155 during *Mtb* infection modulates the expression of a subset of proteins that help establishment of the infection [105], suggesting that cellular miRNAs may provide a mechanism of immune evasion by the pathogen [103,105]. A recent study suggested that exosome-enclosed miRNAs in exhaled breath have been as potential biomarkers for patients with pulmonary diseases including TB [106]. Lyu L et al. in 2019 [37] identified many distinct up regulated and down regulated expressed miRNA and screened top 10 miRNA in the serum of TB patients and LTBI individuals.

Previous reports indicated that miRNAs in serum/plasma as a diagnostic marker for TB infection [107]. The miR-576-3p could differentiate TB patients from health controls. Lyu L and his colleagues [37] identified up-regulation of miR576-3p in TB patients. Furthermore, down-regulation of miR-483-5p and up-regulation of miR-486-5p in TB patients was documented by Lyu L et al. in 2019, which was found to be inconsistent in other published studies [108,109]. Different studies indicated that miRNAs, such as miR-93, miR-29a, miR-378, miR-22, miR-196b, and miR-155, may serve as potential diagnostic markers for TB [37,108,109]. Motto et al. [110] in 2013 identified hsa-let-7e, miR-146a, miR-148a, miR-16, miR-192, miR-193a-5p, miR-25, miR-365, miR-451, miR-532-5p, miR-590-5p, miR-660, miR-885-5p, miR-223, and miR-30e in serum of active pulmonary TB cases. Another study by Liu et al. in 2011 [111] indicated that out of 30 miRNAs 28 miRNAs were upregulated and two were downregulated, with miRNA 144 significantly up regulated in TB cases over healthy controls. Another study on serum of pulmonary TB cases reported that out of 97 miRNA studied 90 were found to be upregulated [112]. In this study, miR-210, miR-432, miR-423-5p, miR-134, miR-144, miR-335, miR-26a, miR-361-5p, miR-889, and miR-576-3p were significantly up regulated among active TB cases. The study suggested that the miRNA levels in the serum of TB patients could potentially detect early pulmonary TB [112]. Liu et al. in 2011 [111], Qi and his colleagues in 2012 [112] and Miotto et al. in 2013 [110] identified that miRNA-144 was significantly up-regulated in TB patients, indicating the potential to use miRNA 144 in TB diagnostic. Furthermore, circulating miRNA support the diagnosis of childhood TB [113]. Latorre et al. in 2015 described novel miRNAs (miR-150, miR-21, miR-29c, and miR-194) in serum for the diagnosis of pulmonary TB patients [114]. A study conducted on patients with latent TB infection (LTBI) indicated that miRNA-889 and tumor necrosis factor (TNF)-like weak (TWEAK) could be potential diagnostic markers for LTBI [115]. Ndzi et al. in 2019 reported the upregulation of miRNAs (miR-146a, miR-155, miR-146a, miR-155, miR-142-5p, miR-423-3p, miR-21-5p, miR-27a-3p, miR-99b, miR-147, miR-223, and let-7i) in bovine TB [116]. The *Mtb* miRNAs extracted from exosomes are summarized in Table 3.

### 3.5. Exosome and Other Lung Diseases

Disease-associated exosomes are studied to improve the diagnostic for various pathologies. Furthermore, their role in the therapeutic targets has been documented [127]. These potential applications have attracted significant interests from both clinicians as well as pharmaceutical industries. However, the precise role of exosomes in disease pathogenesis and their potential for diagnostics and therapy remains to be proven. Exosomes are important indicators of host-derived molecules during infection. Considering their stability in body fluids [59], and their capacity of carrying various microbial components [128,129] acting on target cells (macrophages) or tissues, exosomes provide a great deal of information about the physiological and pathological status of the originating cell via accessing their molecular contents. Exosomal proteins and miRNAs could be a novel therapeutic and diagnostic targets for pulmonary diseases [128]. Analyzing the exosomal proteins and circulating miRNAs profiles could provide information for the lung diseases pathogenesis. A key mechanism of their function, analogous of exosomes, is the fusion of the membrane to the plasma membrane of specific target cells, followed by discharge to the cytoplasm of their luminal cargo containing proteins, RNAs, and DNA [130]. In patients with lung adenocarcinoma exo-DNA was found to be as a diagnostic biomarker [131]. Plasma exosomal protein profiling showed that CD151, TSPAN8, and CD171 were indicative of lung cancer [132]. Serum exosomal miRNA-1290 represents a potential diagnostic and prognostic marker in patients with lung adenocarcinoma [133]. Serum exosomes could be a valuable source of diagnostically important exosomal miR-7977 for the diagnosis of lung cancer [134]. A study conducted by Ueda et al. in 2014 identified CD91 as a lung adenocarcinoma specific antigen on exosomes [135]. The finding was also confirmed in plasma of lung cancer patients by Jakobsen KR and his research team in 2015 [136]. Serum Exo-miRNAs such as miRNA-34, let7, miRNA-21, and miRNA-1792 have been reported as lung cancer regulators [137,138,139]. Plasma exosomal markers such as NY-ESO-1, miRNA could be applicable in the diagnostics of lung tumor [132]. In addition, pleural effusion exosomal miRNAs, including miRNA-483-5p, miRNA-375, and miRNA-429, were identified as promising lung cancer biomarker candidates [117]. A recent report suggests that endothelial EVs proteins CD31+, CD66+, and CD235ab+ might be chronic obstructive pulmonary diseases (COPD) specific markers [140]. Furthermore, in patients with COPD increased concentrations of CD144, CD31, and CD62E EVs have been reported [141], indicating that CD144, CD31, and CD62E could be a surrogate diagnostic marker for COPD exacerbation. A study conducted by Ghiot J et al. in 2019 [142] reported that miRNA-629, miRNA-223-3p, and miRNA-142-3p largely affect lung inflammation and contribute to asthma progression. Exosomes, which are enriched for caspase-3, mediate the activation of alveolar macrophages and the propagation of the inflammatory responses resulting in lung injury [143]. Exosomal miRNAs, miRNA7 and miRNA125b play an important role in the development of idiopathic pulmonary fibriosis [144]. 

## 4. Conclusions

A variety of exosome isolation methods suitable for TB patient-derived samples and cell culture supernatants has been reported. Differential centrifugation techniques remain common and gold standard; however, other methods, such as immunological separation and precipitation show promising results and can be effectively applied both in *Mtb* exosome research. Secreted exosomes could be candidates for various clinical applications in TB. Analysis of *Mtb* exosomal content has revealed a variety of proteins and miRNAs, which may be releavant for TB pathogenesis. Exosomes are endowed with several advantages including the stable conformational conditions for the proteins, the presence and molecular distribution in body fluids and possibility to reach distal organs and a more efficient association of antigen with target cells [145]. In addition, the identification of easily measured and accurate diagnostic exosomal TB markers will have a significant impact on vaccine studies. The data presented here supports the functional and diagnostic potential of exosomal protein and/or miRNAs in TB. Thus, there is potential to use the *Mtb* derived exosomal molecules in TB diagnostic and vaccine discovery, indicating that the extraction of exosomes from biological fluids is an essential footstep in exosome research.

## Figures and Tables

**Figure 1 jcm-10-00145-f001:**
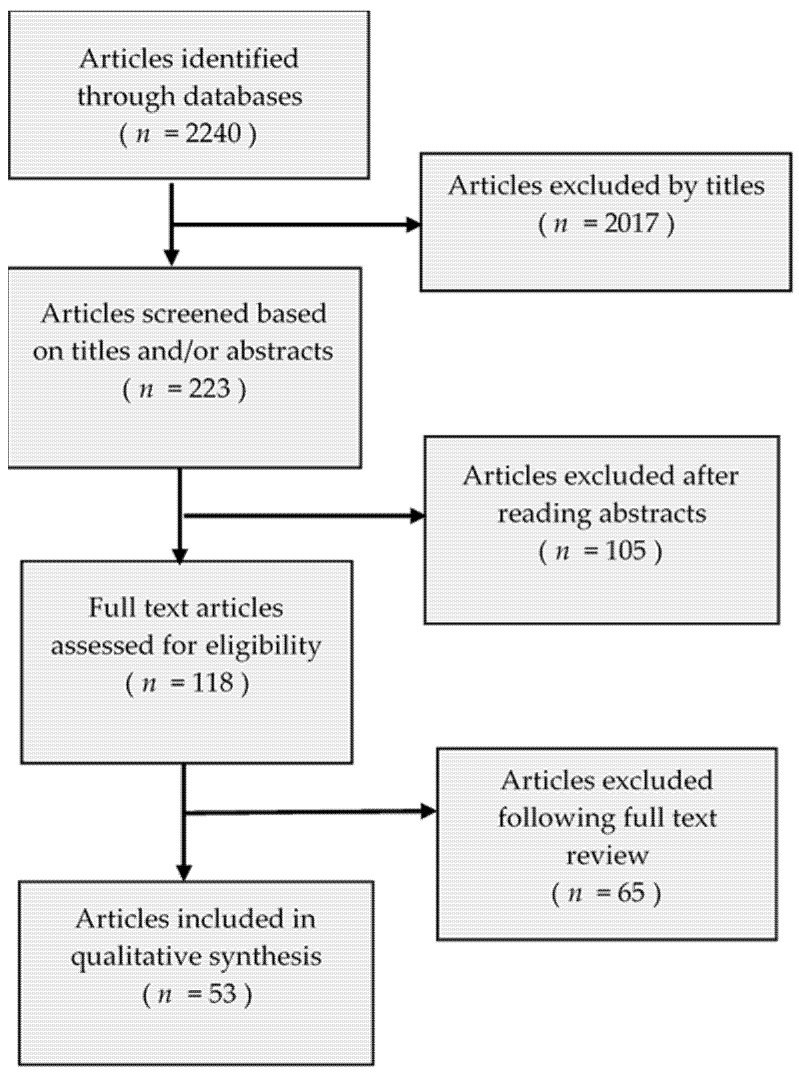
Flow diagram showing literature review.

**Figure 2 jcm-10-00145-f002:**
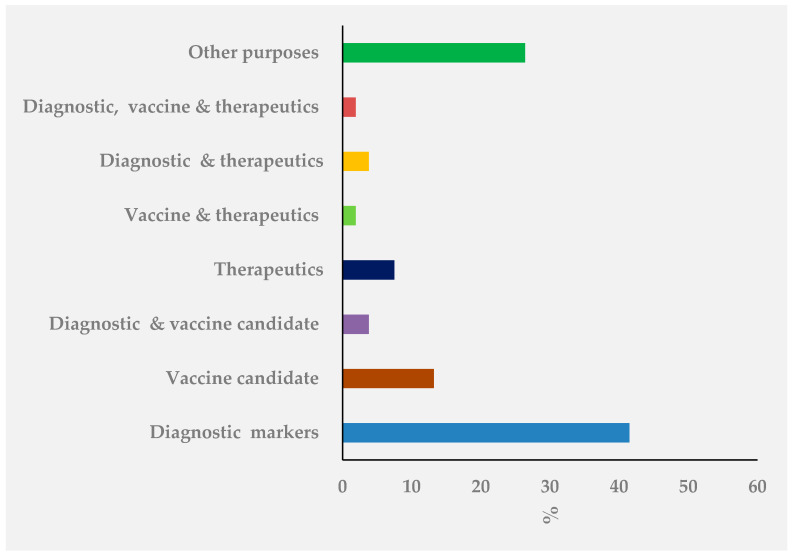
Publication frequencies of studies by conclusion and clinical significance of exosome research in tuberculosis (TB) based on PubMed/Medline and Google scholar search, 2020.

**Figure 3 jcm-10-00145-f003:**
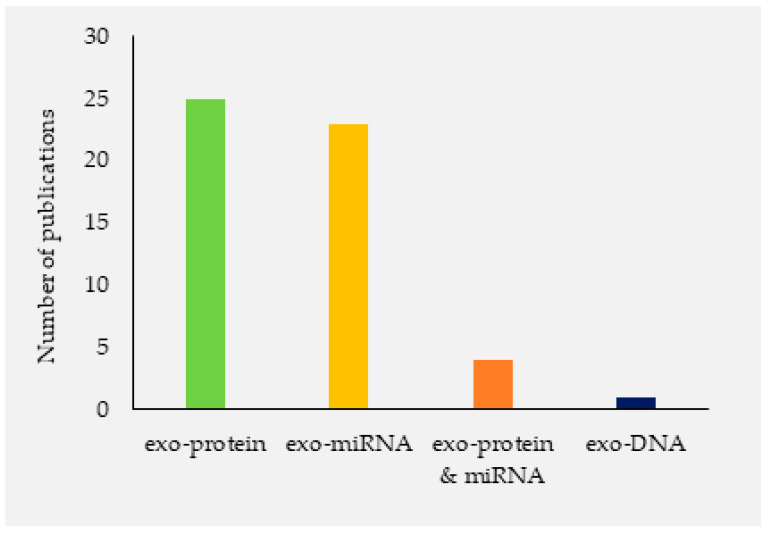
Publication frequencies of studies investigating *Mycobacterium tuberculosis* (*Mtb)* derived molecular content of exosomes based on PubMed/Medline and Google scholar search, 2020.

**Figure 4 jcm-10-00145-f004:**
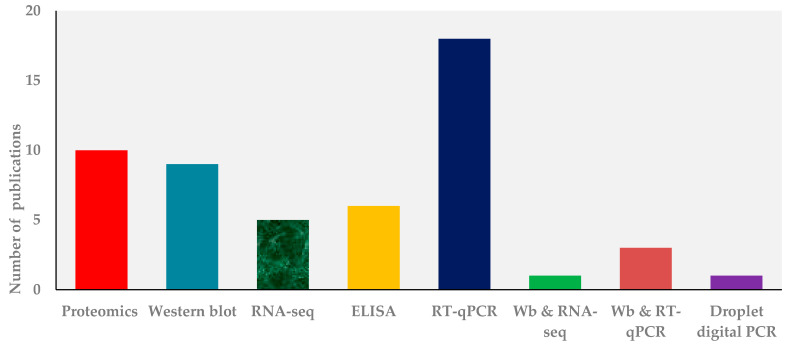
Publication frequencies of techniques employed in TB exosome research for Mtb-derived molecular content of exosomes analysis based on PubMed/Medline and Google scholar search, 2020, Wb, western blot.

**Figure 5 jcm-10-00145-f005:**
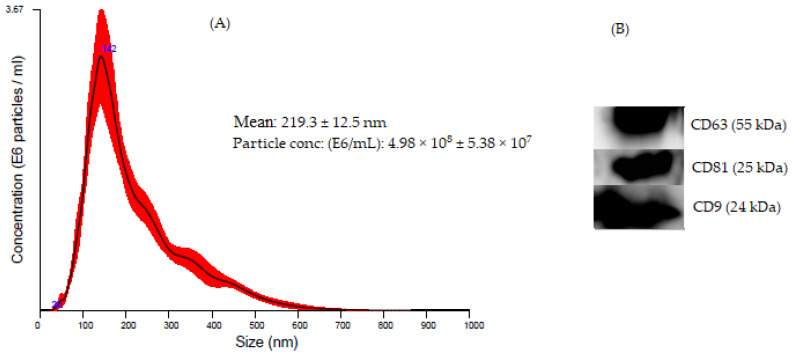
Characterization of exosomes extracted from THP-1 cell culture supernatants (**A**) Nano sight NS300. A concentration of 1E9/mL vesicle suspensions was analyzed using Nano sight NS300 (Nano sight Ltd, Amesbury, UK) equipped with 450 nm laser to determine the size and concentration of particles isolated according to the manufacturer’s protocols. Each experimental run was performed in triplicate (*n* = 3) (each capture 60 s with a frame rate of 25 frames/s and a camera level of 9). The y-axis shows the number of particles/mL and the x-axis shows the size distribution of particles in nm. The concentration is indicated in particles/mL and the particle size (mean ± SD nm) are shown, and (**B**) Western blot analysis of THP-1 derived exosomes with anti-CD9, anti-CD81, and anti-CD63. Exosome pellet was directly lysed in lysis buffer (25 mM Tris-HCL (pH 7.6); 150 mM NaCl, 1%NP-40,1% Sodium deoxycholate, 0.1% SDS: Invitrogen, Thermofisher Scientific Inc, Dreieich, Germany) for 15 min at 4 °C. Lysate exosomes (30 µL) were separated on 12% SDS-PAGE under a non-reducing condition and transferred into PVDF-membranes, incubated overnight with the specific primary antibodies (CD9, CD81, and CD63), followed by HRP-conjugated secondary antibodies and detected using a super signal West Femto substrate (Invitrogen, Thermofisher Scientific Inc, Dreieich, Germany) (unpublished data).

**Table 1 jcm-10-00145-t001:** Some host exosomal markers by *Mtb* infected cells based on PubMed/Medline and Google scholar search, 2020.

Year	Isolation Technique	Material	Markers	References
2015	Sucrose gradient density centrifugation	Cell culture	CD9, CD63, Alix, LAMP-1and GAPDH	[47]
2010	UC	Cell culture	LAMP-1	[63]
2016	ExoQuick-TC	Cell culture	CD63, CD81, RAb5B, HSP70	[71]
2007	UC, Sucrose gradient density centrifugation	Cell culture	HSP70, MHC-I, II, CD86	[74]
2019	ExoQuick-TC Ultrafiltration	Cell culture Serum	CD63, Alix, RAb5b, LAMP-2	[76]
2017	TEIR	Serum	CD81	[77]
2020	ExoQuick-TC	Serum	CD63, CD9	[78]
2019	UC	Cell culture	CD63	[79]
2017	UC	Cell culture	CD9, CD63	[80]
2015	ExoQuick_TC	Cell culture	LAMP-1, CD40, CD80, CD81, CD86, CD195	[81]
2015	UC	Cell culture	TSG101, LAMP-1	[82]
2010	UC	Cell culture	MHC-II, CD86, Rab 7, ICAM-I	[83]

UC, ultracentrifugation, TEIR, Total Exosome Isolation reagent.

**Table 2 jcm-10-00145-t002:** *Mtb* molecules extracted from exosomes based on PubMed/Medline and Google scholar search, 2020.

Size (nm)	Method	Exosomal Molecules	Sources	References
NI	Ultrafiltration	LpqN, LprA, LprF, LprG, LpqH, MPB83, FEIII, LpqL, LppX, LppZ, PBP-1, PSTS3, phoS	Culture	[39]
NI	Ultrafiltration and UC	pstS1, glnA1, garA, Dank, fba, icd2, ctpD, sahH	Serum	[41]
85–150 nm	Sucrose density gradient	lipoproteins (LpqH and LprG) and lipoglycans (LAM and LM)	culture	[47]
30–50 nm	TEIR for urine	CFP10, lipoarabinomannan	Urine	[50]
30–50 nm	Filtration and UC	KatG, GlnA, SodA, GroES, CFP10, Ag85,19kDaLpqH	culture	[63]
30–50 nm	ExoQuick-TC	60 kDa heat shock protein, mitochondrial, Moesin, 60 S acidic ribosomal protein P0, 78 kDa glucose-regulated protein, ATP-dependent RNA helicase A, Lamin-B1, Heat shock cognate 71 kDa protein	culture	[71]
NI	UC	Ag85 complex	Culture	[72]
NI	Centrifugation	HSP16	Serum	[73]
50–150 nm	UC	LAM and 19-kDa lipoprotein	culture	[74]
30–50 nm	ExoQuick-TC, UC	KYAT3, SERPINA1, HP, and APOC3	Serum	[78]
NI	UC	LAM, LM	Cell culture	[80]
30–100 nm	ExoQuick	Ag85C, MPT51, SodA, MPT63, ESAT-6	Culture	[81]
NI	Sucrose gradient	KatG, HspX, and GroES	culture	[82]
NI	ExoQuick	AcpM, Ag85a,b,c, Apa, BfrB, Cfp1, DnaK, Fba, GlcB, GroES, HspX, Icd2, KatG, Mpt64, SahH, PpiA	Serum	[86]
NI	ExoQuick	Ag85a,b,c, GlnA, Mpt32, Mpt 64, BfrB, HSPX, Cfp2, GLCB, Cfp10, Dank, GroES	Serum	[88]
NI	Gradient filtration and centrifugation	Ag85 A, B, C, MPT64 and LprG	Serum	[89]
50–150 nm	UC	19-kDa Lipoprotein	culture	[101]
20–200 nm	Methanol precipitation	SodB, EsxN, LppX, PstS1, LpqH, KatG, GlnA1, Apa, FbpA, FadA3, Mtc28, AcpM, Fba, and Prs	Culture	[102]

UC, Ultracentrifugation, TEIR, Total Exosome Isolation Reagent, LAM, lipoarabinomannan, LM, lipomannan, NI, Not Indicated.

**Table 3 jcm-10-00145-t003:** *Mtb* miRNA extracted from exosomes based on PubMed/Medline and Google scholar search, 2020.

Exosomal miRNAs/DNA	Sources	References
let-7e-5p, let-7d-5p, miR-450a-5p, miR-140-5p miR-1246, miR-2110, miR-370-3P, miR-28-3p miR-193b-5p	Serum	[37]
miR-1224, miR-1293, miR-425, miR-4467, miR-4732, miR-484, miR-5094, miR-6848,miR-6849, miR-4488 miR-96	Culture	[77]
miR-20b-5p	Culture	[79]
miR-191, miR-20b, miR-26a, miR-106a, let-7c, miR-20a, miR-486, miR-3128, miR-1468, miR-3201, miR-8084	Plasma	[91]
mir-149-3p, mir181c-5p, mir1839-3p, mir151-3p, mir241-3p, mir292-3p, mir3107-5p, mir344i, mir486-5p, mir486-3p, mir434-5p, mir714, mir877-3p, mir759, mir713	Culture	[103]
let-7e, let-7f, miR-10a, miR-21, miR-26a, miR-99a, miR-140-3p, miR-150, miR-181a, miR-320, miR-339-5p, miR-425, and miR-582-5p	Serum	[111]
miR-210, miR-432, miR-423-5p, miR-134, miR-144, miR-335, miR-26a, miR-361-5p, miR-889 and miR-576-3p	Serum	[112]
miR-148a-3p, miR-451a, and miR-150-5p	Pleural effusion	[117]
miR-18a	Culture	[118]
miR-484, miR-425, and miR-96	Serum	[119]
miR-155, miR-150 and miR-193	Biopsies	[120]
miR-126-3p, miR-130a-3p, miR-151a-3p, miR-199a-5p, miR-642a-3p, and miR-4299	Serum	[121]
miR-21, miR-223, miR-302, miR-424, miR-451 and miR-486-5p, miR-144, miR-365, miR-133a, miR-424	Blood	[122]
miR-340-5p miR-451a miR-32-5p miR-27a-3p miR-29a miR-29b	Blood	[123]
miR-424	Blood	[124]
miR-3615, miR-4516, and miR-378i	Biopsy	[125]
ExoDNA	Sputum	[126]

## Data Availability

Data is contained within the article.

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
