# Peer review of "Composition and Clinical Significance of Exosomes in Tuberculosis: A Systematic Literature Review"

_jcm, 2021, doi:10.3390/jcm10010145_

Round 1

Reviewer 1 Report

This is a very well organized and well written review with focus on Extracellular Vesicles (EVs) in TB with implications in other lung diseases. The authors have systematically reviewed the current literature in the field with focus on methods for EV isolations and characterization including limitations as well as use of  components of EVs as potential diagnostic markers and for future vaccine development.

Author Response

 Word document is attached to reviewer 1

Reviewer 2 Report

The authors provided a well-documented overview of Ev implications in infectious diseases and, specifically related to tuberculosis. In addition to this, this article provides an interesting point of view regarding the advantages and disadvantages of EV-like particles in TB diagnostics and vaccine innovations. I already suggested adding some more recent references (PMID: 29505299 and PMID: 29029605 are just as an example) but I didn't found them in the revised version.
I look forward to receiving the updated version of the paper.

Author Response

Attached here with the word document to R2

Reviewer 3 Report

The article aims to summarize the current knowledge regarding the relationship between exosomes and TB. Unfortunately, the article cannot be published in this form and requires significant changes.

  1. The purpose of the work is not clearly defined.
  2. 2. In the introduction, the authors should define/assume the role that exosomes can play in TB. The description of the body fluids in which exosomes are present is not necessary - it should be simplified (they are present in almost every body fluid: https://doi.org/10.3390/cancers12103019). It can be noted here that in the course of infectious diseases, they can modulate the inflammatory response (https://doi.org/10.3389/fimmu.2018.02723).
    3. It is not clear what kind of review the authors want to present - systematic or narrative? It is not either of them at the moment.
    If the authors want to present a systematic review (and it looks like it) - please do so according to the rules described in PRISMA.
    4. The authors should focus more on describing the function of exosomes in TB - the JCM is a clinical journal and this section may be of most interest to practitioners.
    5. Figures are of poor quality.

Author Response

Attached here with the word document to R3

Reviewer 4 Report

The review is overall well written and follows a logical thread. I took note of the comments of previous reviewers noting that the authors answered the points comprehensively.

In any case, when it comes to the subject of exosomes one cannot, nowadays, not refer to their fundamental role in liquid biopsy as a premise to the discussion of the subject of your review.

In this regard, I recommend adding some period between lines 50-55, perhaps citing some recent work about exosomes in monitoring and liquid biopsy such as PMID: 32759810

Moreover, minor spell and English form check is required.
Below, some comments about.

1 - line 195: "diffrent" ;

2 - line 107: "superntants" and so on;

3 - it would be better to increase the size of the text in figure 2;

4 - line 110: the major drawbacks;

5 - line 131: "based on the interaction of surface receptors (proteins) of exosomes and antibodies": the interaction between exosomes surface receptors and antibodies;

6 - Overall, a punctuation check is required: for example in line 244 the spacing must go after the comma and not before (and so on). 

Best regards

Author Response

Attached here with the word document to R4

Round 2

Reviewer 3 Report

The authors have addressed all the comments of the reviewer and revised the manuscript accordingly. 

Congratulations and Happy New Year!

This manuscript is a resubmission of an earlier submission. The following is a list of the peer review reports and author responses from that submission.

Round 1

Reviewer 1 Report

This is a very well conceived and rationalized comprehensive review focused on the isolation methods and characterization of the composition of mycobacterial extracellular Vesicles/Exosomes (proteome and miRNA analysis) from cultures and biological fluids from patients. The advantages and limitations of the techniques used for  the isolation of the vesicles were clearly described. The authors have used original and current research articles (based on their inclusion criteria) and concluded the potential of components of exosomes in TB diagnostics and vaccine discovery. 

The review article is well organized and written clearly.

Author Response

Kindly ,  Point by Point PDF file  is attached

Reviewer 2 Report

The authors provided a well-documented overview of Ev implications in infectious diseases and, specifically related to tuberculosis. In addition to this, this article provides an interesting point of view regarding the advantages and disadvantages of EV-like particles in TB diagnostics and vaccine innovations. The field of research focused on exosomes is in continuous evolution and even if the article is well written, the references could be updated with more recent works  (PMID: 29505299 and PMID: 29029605 are just as an example).
I look forward to receiving the updated version of the paper.

Author Response

Kindly, Point by Point Response to reviwers comment has been attached

Reviewer 3 Report

The review covers composition and clinical significance of exosomes in tuberculosis. Unfortunately, on the topic of tuberculosis information is only from 11 to 17 pages, half of them are tables.

The abstract contains semantic repetitions (the last three sentences). Abbreviation for extracellular vesicles is EVs, not Evs! Abbreviation PM in Table 1 not disclosed.

There are inconsistencies in the size of the exosomes in the text at page 2 (Exosomes are small-sized particles (30-150 nm) that are produced by the inward budding of early endosomes [4].) and in the table 1, although reference [4] is used in both cases. Refs [8]&[17] are inappropriate, but necessary references are missing elsewhere. For example, in sentence “Exosomes has been identified [8?????] in most biological fluids including plasma [?], serum[?], milk[?], urine and saliva[?], may transfer information to the nearby cells and hence have an important role in cell-to-cell communication[?].”

In “3.1 Finding“ there are no conclusions on Figure 2. Why then give it?

The section “3.1.1. Isolation of exosomes” with Table 2 are highly controversial! For example, size exclusion chromatography is a not “golden standard” of exosomes isolation. Such dubious statements are present about the advantages and disadvantages of each of the listed methods.

The Fig 3 has no semantic meaning

ExoCarta is database, and not method! (section 3.1.2.)

No  conclusions on the section 3.1.2. and no conclusions on Figure 4. Why then give it?

And etc….

An interesting title does not match the content of the article. Most text (64%) have nothing to do with the stated topic, but retells with dubious conclusions already published known data many times.

Author Response

Kindly, Point by Point response has been uploaded in PDf file

Round 2

Reviewer 3 Report

According to the PubMed database on November 10, 4909 reviews on the topic of extracellular vesicles were published; a huge number of them are reviews devoted to methods of isolation, characterization and functions of vesicles in the body. Your review would greatly benefit from focusing specifically on vesicles in tuberculosis. Thus, the sections 1.1, 3.1.1, 3.1.2, 3.1.3. need to be rewritten specifically regarding publications on vesicles in tuberculosis. The section 3.1.4 need focus to lung diseases (not ALL diseases at world).

Some irrelevant refs were left without correcting. For example:

Exosomes modulate physiological processes such as lactation, inflammation, cell proliferation, immune responses, and neuronal function [15–20], and have been implicated in disease progression, for instance in cancer liver diseases and neurodegenerative conditions [21–23].

Ref 23: Kang, H.; Kim, J.; Park, J. Methods to isolate extracellular vesicles for diagnosis. Micro and  Nano Syst Lett 2017, 5, doi:10.1186/s40486-017-0049-7.

Currently, researchers in the field are busy studying tiny ‘messengers’ between the cells called exosomes, these nano-sized vesicles enclosed by a membrane of lipids, transfer molecules from cell to cell [8].

Ref 8: Lobb, R.J.; Becker, M.; Wen, S.W.; Wong, C.S.F.; Wiegmans, A.P.; Leimgruber, A.; Möller, A. Optimized exosome isolation protocol for cell culture supernatant and human plasma. J. Extracell. Vesicles 2015, 4, 27031, doi:10.3402/jev.v4.27031.

Etc…

Author Response

Response to reviewer 3 is attached here!!!
